# Absolute Quantitation of Serum Antibody Reactivity Using the Richards Growth Model for Antigen Microspot Titration

**DOI:** 10.3390/s22103962

**Published:** 2022-05-23

**Authors:** Krisztián Papp, Ágnes Kovács, Anita Orosz, Zoltán Hérincs, Judit Randek, Károly Liliom, Tamás Pfeil, József Prechl

**Affiliations:** 1R&D Laboratory, Diagnosticum Zrt, 1047 Budapest, Hungary; pkrisz5@gmail.com (K.P.); herincsz@diagnosticum.hu (Z.H.); 2Department of Applied Analysis and Computational Mathematics, Eötvös Loránd University, 1117 Budapest, Hungary; agnes.kovacs@ttk.elte.hu (Á.K.); tamas.pfeil@ttk.elte.hu (T.P.); 3Department of Immunology, Eötvös Loránd University, 1117 Budapest, Hungary; orosz.anita@med.semmelweis-univ.hu; 4Budapest University of Technology and Economics, 1111 Budapest, Hungary; randek.judit@hotmail.com; 5Department of Biophysics and Radiation Biology, Semmelweis University, 1085 Budapest, Hungary; karoly.liliom.mta@gmail.com; 6ELKH-ELTE Numerical Analysis and Large Networks Research Group, 1117 Budapest, Hungary

**Keywords:** polyclonal, antibody, quantitative, serology, antigen, immune response, chemical potential, activity coefficient, microarray, Richards curve

## Abstract

In spite of its pivotal role in the characterization of humoral immunity, there is no accepted method for the absolute quantitation of antigen-specific serum antibodies. We devised a novel method to quantify polyclonal antibody reactivity, which exploits protein microspot assays and employs a novel analytical approach. Microarrays with a density series of disease-specific antigens were treated with different serum dilutions and developed for IgG and IgA binding. By fitting the binding data of both dilution series to a product of two generalized logistic functions, we obtained estimates of antibody reactivity of two immunoglobulin classes simultaneously. These estimates are the antigen concentrations required for reaching the inflection point of thermodynamic activity coefficient of antibodies and the limiting activity coefficient of antigen. By providing universal chemical units, this approach may improve the standardization of serological testing, the quality control of antibodies and the quantitative mapping of the antibody–antigen interaction space.

## 1. Introduction

Antibodies are glycoproteins belonging to the immunoglobulin protein superfamily, produced by B cells of the adaptive immune system. Antibodies in an organism are structurally highly heterogeneous owing to the gene-recombination-mediated diversity of the adaptive immune response and the clonal nature of B-cell expansions [1,2]. A single given B cell and its progeny of plasmablasts and plasma cells produce antibody molecules with a given target specificity and binding affinity, determined by the fit of the rearranged heavy and light chain variable regions. Millions of B cells carrying different rearranged heavy and light chain variable regions generate a network of diverse antibodies present in an individual [3]. A humoral immune response against a particular antigen involves the evolution of antibody-producing plasmablast and plasma cell clones with various isotypes and affinities, the latter properties also being characteristic for the polyclonal response in addition to target specificity. The affinity of an antibody molecule towards its target antigen is an important characteristic both from the biochemical and immunological point of view. Biochemically, it reflects the strength of interactions between the two binding partners [4]. Immunologically, it is closely correlated with effector functions such as neutralization efficiency and complement activation [5]. The isotype of the antibody molecule is the other key determinant of an effector function. Understanding the properties of serum antibodies generated against particular antigenic targets is the goal of serological measurements, which aim at the laboratory diagnosis of infectious, allergic and autoimmune diseases [6,7,8]. Evidently, medical laboratory diagnostics requires the standardization of serological measurements.

Standardization of the measurement of antibody reactivity has always been a difficult issue [9], and in spite of much effort, it has not been fully resolved [10]. Among the analytical issues is the fact that the quantitation of a heterogeneous population of antibodies would require a heterogenous reference material. Currently, this is achieved by the utilization of affinity-purified polyclonal serum pools or pools of monoclonal antibodies. These reagents, along with the standardization and harmonization of measurement protocols, led to current best laboratory practices, which provide results in international units. However, the quantitative comparison of different assays with different units is not meaningful; therefore, the systems-level quantitative representation of antibody reactivity is currently impossible.

Determination of antibody affinity has always been considered of key importance for the characterization of humoral immunity. From the immunochemical and biophysical point of view, the techniques employed range from equilibrium dialysis [11], precipitation assays using radiolabeling [11] and use of denaturants to disrupting binding forces [12], setting up competition assays [13] and varying Ag density in enzyme-linked immunosorbent-assays (ELISA) [14]. A non-competitive ELISA [15] for the measurement of monoclonal antibody affinity that relies on the determination of Ab binding at different antigen coating densities shares some conceptual basis with our assay; however it requires the use of known concentrations of monoclonal antibodies. Alternatively, equilibrium binding constants can be obtained from kinetics measurements by surface plasmon resonance analysis [16], biolayer interferometry [17], microscale thermophoresis [18] and similar technologies. While these measurements are often regarded as the golden standard of affinity determination, the instrumentation and data-intensive nature of measurement may not be optimal for routine diagnostic use. Additionally, label-free methods are not easily adaptable for the selective measurement of antibody classes, a must for immunodiagnostics.

Serum antibody titration is a simple and widely used way of estimating antibody reactivity, wherein the sample is serially diluted and the highest dilution that gives a signal reliably discernible from the background, or the dilution resulting in half signal, are regarded as the titer for endpoint and midpoint titration, respectively. Such titers may provide fast and cost-effective semi-quantitative results but are not suitable for the real quantitation of serum antibodies because we are changing an unknown variable when we decrease antibody concentration by serum dilution.

The unique properties of microspot immunoassays were described by Ekins [19], who propagated their use for capture immunoassays [20]. The general idea is that owing to the negligible amount of molecules present in the microspot probe compared to the amount of analyte in the tested solution, these assays are mass independent [20]: interactions are governed by concentration and affinity only. When affinity is constant and homogenous, as in a capture immunoassay utilizing monoclonal antibody, the equilibrium density of analytes bound to the surface by capture antibodies is determined solely by its concentration. Here, we propose that mass independence of microspot immunoassays in fact requires an analytical approach that is different from conventional calculations based on the law of mass action [15,21], which applies to reactions in solutions. We suggest the use of concepts and terminology of physical chemistry, which are readily applicable to reactions at a solid–liquid interface [22,23].

In this paper, we present a microspot-based approach that allows the estimation of apparent chemical potential of distinct antibody isotypes in human serum without any prior purification steps. The measurement relies on the simultaneous titration of antigen surface density and serum dilution and on an improved mathematical model of fitting. Most importantly, by changing a known variable of the system, antigen concentration, the technology allows the absolute quantification of antigen-specific reactivity of polyclonal serum antibodies, allowing for quantitative comparison of different samples, isotypes or antigens.

## 2. Materials and Methods

### 2.1. Microarray Production and Measurements

Experiments were carried out on hydrogel-coated glass slides (Nexterion Slide H, Schott, Jena, Germany) by using a BioOdyssey Calligrapher MiniArrayer (Bio-Rad Laboratories, Hercules, CA, USA). Different peptide dilutions in 1% dimethyl sulfoxide, 2% glycerol, 0.001% Tween 20, were spotted in quintuplicates of half serial dilution in 7 steps starting from 200 μM, with a layout shown in Figure 1. Slides were dried for 1 h at 37 °C then soaked in 0.1 M Tris buffer (pH = 8.0) for 1 h at 37 °C in order to inactivate reactive residues on the surface. Once prepared, slides were stored in sealed non-transparent bags at 4 °C. The fusion peptide GLN1-2F was based on the deamidated gliadin sequence motifs LQPFPQPELPYPQPQ and PLQPEQPFP and was synthesized by Bio Basic Canada Inc (Markham, ON, Canada). Unless otherwise stated all reagents were from Sigma-Aldrich (St. Louis, MI, USA).

Frozen serum samples collected for studies with contract numbers 24933-6/2018/EKU and 24973-1/2012/EKU (658/PI/2012.) were used for the experiments. We selected sera that tested positive for transglutaminase antibodies with a commercially available ELISA kit (re-hu-TG-IgA test, Diagnosticum, Budapest, Hungary). Those studies were aimed at developing serological assays for gluten sensitivity and autoimmune diseases, respectively, and all procedures were in accordance with the ethical standards of the responsible committee on human experimentation and with the Helsinki declaration of 1975, as revised in 2008.

### 2.2. Sample Handling and Signal Detection

The basic protocol was the following: dried arrays were rehydrated in 110 µL PBS (3 × 5 min) before using, then sub-arrays were incubated in 70 μL diluted sample at 37 °C for 1 h. Sample dilutions, as indicated in Figure 2, were carried out in PBS-BSA-Tween (0.05% Tween 20, 2% BSA, PBS). Serum-treated slides were washed in 0.05% Tween-PBS, then incubated at room temperature for 30 min with fluorescently labelled antibodies that were diluted in the blocking buffer (0.05% Tween 20, 2% BSA, PBS). Fluorescently labelled secondary antibodies, Fc γ fragment-specific DyLight 649-conjugated AffinityPure F(ab’)2 fragment goat anti-human IgG (Ref.: 109-496-008, Jackson Immunoresearch, West Grove, PA, USA); DyLight 549-conjugated AffinityPure F(ab’)2 fragment goat anti-human-IgA (Jackson Immunoresearch, Ref.: 109-506-011) were used as a mixture at 1:2000 dilution. All secondary antibody dilutions were prepared in 0.05% Tween 20, 0.5% BSA in PBS. Chips were washed again and following drying, slides were scanned using an Axon GenePix 4300A microarray scanner (Molecular Devices, San José, CA, USA).

### 2.3. Measurements with 14D5 Monoclonal Antibody

The above-described basic protocol was altered for the assays with monoclonal antibody against deamidated gliadin (14D5, Ref.:36729, Abcam, Cambridge, UK). First, slides were rehydrated in PBS (3 × 5 min) then blocked in blocking buffer (0.05% Tween 20, 2% BSA, PBS) for 30 min at 37 °C. A dilution series was prepared from the monoclonal antibody starting from 200 μg/mL by half in 10 steps. Detection was carried out by goat anti-mouse IgG2a-Alexa488 (Ref.: A21131, Life Technologies, Carlsbad, CA, USA) diluted 1:4000. Incubations were at room temperature for 30 min.

### 2.4. Analysis of the Microarray Data

Images of the slides were analyzed with GenePix Pro 6.0 software after visual inspection. Spots were recognized by the program, then gpr files containing the spot coordinates for individual spots were created. Then, R (version 3.5.2), a statistical programming environment, was used to re-analyze 16-bit tiff images by using these coordinates in order to align the spots. Relative fluorescence intensity (RFI) medians were calculated for each spot using 80% of the diameter of the circle shapes that were previously adjusted on the features by the software. The reduction in the diameter of the microspots served to exclude diffuse borders with lower signals from the analysis. In the next step, means of the 5 parallel spots were taken and values of the least concentration antigen spots were subtracted of all RFI values, thus providing the final signal intensities. The reason for using spots with least antigen concentration as background was partly empirical, as we found signals at this 3 μM antigen concentration to correspond to serum-specific background signals, and partly theoretical, because we used curve-shaped parameters rather than fluorescent intensity values for characterization of binding.

### 2.5. Fitting of Binding Curves

In the indirect assay, we use a linear model for polyclonal reactions [24]. We take into consideration that a bound antibody inhibits nearby free antigens from forming complexes with other antibodies [25,26]. This inhibition makes the concentration of immune complexes a logistic function of the logarithm of the total antigen concentration:(1)$$\left[\mathrm{AbAg}\right]=\left[\mathrm{Ab}\right]*\left[\mathrm{Ag}\right]/\left(K_{D}+\left[\mathrm{Ab}\right]+D\;*\left[\mathrm{Ab}\right]*\left[\mathrm{Ag}\right]\right)$$
where D is a positive constant (for details see Appendix A).

It is the nature of our measurement system that the logarithm fluorescent intensity is a linear function of the logarithm concentration of AbAg complexes in the range studied. By using these, we obtain that the logarithm of fluorescent intensity is a Richards function of the logarithm of total antigen concentration:(2)$$R\left(x\right)=A\;*\;{\left(1+\left(d-1\right){\;*\;e}^{-k\left(x-x_{i}\right)}\right)}^{\frac{1}{1-d}}$$
with k = 1, where A is the total antibody concentration [Ab] (limit of function R(x) at infinity), x_i_ is the inflection point and d is the asymmetry parameter (for details see Appendix A).

The upper limit of the fluorescent intensity A depends on the serum dilution. Formula (1) also implies that the concentration of immune complexes is a logistic function of the logarithm of the total antibody concentration ln[Ab] with growth rate 1, i.e., a logistic function of −ln(***z***) with growth rate 1, where z is the serum dilution factor. Due to the linear dependence of the logarithm fluorescent intensity on the logarithm printed Ag concentration, the upper limit of the logarithm fluorescent intensity is a Richards function of the negative logarithm serum dilution z with growth rate 1.

Therefore, the fluorescent signal intensity against the logarithm total antigen concentration x and the negative logarithm of serum dilution, −ln(z), is the product of two generalized logistic functions of the following form:(3)$$R_{2}\left(x,z\right)=C\;*\;{\left(B+z\right)}^{-m}*{\left(1+\left(d-1\right){\;*\;e}^{-\left(x-x_{i}\right)}\right)}^{\frac{1}{1-d}}$$

The logarithm transformation converts the proportional variance pattern to a constant variance pattern, and thus the conversion makes the transformed data more suitable for fitting the model. The above multiplicative relationship now changes to an additive one in the form of:(4)$${\mathrm{lnR}}_{2}\left(x,z\right)=\ln\;\left(C\right)-m*\ln\;\left(B+z\right)+\frac{1}{1-d}*\ln\left(1+\left(d-1\right){\;*\;e}^{-\left(x-x_{i}\right)}\right)$$

This generalized logistic model on the log-log scale was fit to the data, and parameter estimates and 95% confidence intervals were obtained for each serum and both isotypes. All analyses were carried out using the R software (version 3.5.2). 

Nonlinear least squares estimates for the model parameters were calculated using the Gauss–Newton algorithm of the nls function from the statistical software package R (version 3.5.2). The 95% confidence intervals generated for the model parameters were based on the profile likelihood technique. Figures are representative of at least 3 independent experiments.

## 3. Results

### 3.1. Experimental Setup and Properties of the Measurement System

The experimental system we use is characteristic of protein microarray technology with some key differences to traditional indirect ELISA. The hydrogel-coated solid surface used as the antigen adsorbent has high binding capacity, which results in a dose-dependent adsorptive binding over a wide concentration range. Therefore, the concentration of antigen in the solution used for bio-printing (spotting of antigen) is expected to show a linear relationship with the surface density of the antigen on the microarray over a correspondingly wide range. Antibodies that bind to the spotted antigens are detected by fluorescently labelled secondary antibodies. Owing to the properties of fluorescent measurement, bound antibodies can be detected over a wide range as well. To allow for the analysis and visualization of signals spreading over several orders of magnitude, the logarithm of both antigen concentrations and fluorescent signal intensities is used.

To establish a distribution of immune complex concentrations as a function of antigen density, we employed two different complementary strategies (Figure 1). First, increasing concentrations of antigen [Ag] were printed on a solid surface as microspots. Owing to the negligible amount of antigen in a microspot compared to the amount of antibody in the reaction solution, these measurement conditions are called mass-independent or ambient analyte immunoassay [19,20]. Antigen microspots therefore probe antibody reactivity in the sample without significantly altering composition of the sample. Second, dilutions of the serum samples of interest were prepared to vary the concentration of antibody–antigen complexes [AbAg] in the measurement range and to examine the relationship between antibody concentration [Ab] and [AbAg]. Even though the absolute free antibody concentration [Ab] in the tested serum is not known, the relative concentrations of the dilution series can be used to follow [Ab] effects by mathematical curve fitting. Dilutions of serum samples were applied to the series of antigen density microspots and incubated to bring the system into equilibrium.

The resulting measurement system examines the effect of changes in relative [Ab] and [Ag] on the extent of Ab binding. The obtained experimental data are fitted with a growth function to derive values of the parameters that characterize antibody reactivity in the sample (Figure 1).

### 3.2. Curve Fitting

The formation of AbAg complexes and the extent of antibody binding under equilibrium in our assay can be interpreted as the indicator of thermodynamic Ab activity in the microspots. The equilibrium concentration of AbAg complexes is a logistic function of ln[Ag], and the logarithm of the fluorescent intensity is a linear function of ln[AbAg]; therefore the growth in fluorescent intensity can be described by the generalized logistic function [27,28] or Richards function R(x) of x = ln[Ag], an extended form of the function most frequently used for immunoassays. We use the parametrization shown in Formula (2).

The capacity of the system is determined by the availability of antibody, [Ab]. While the exact value is unknown, relative values corresponding to steps in dilution series can be used in the fitting procedure. In our assay, exponential growth in [AbAg] is allowed by the provision of exponentially increasing [Ag] in a series of microspots. We assume that antigen titration results in a binding curve following a generalized logistic curve with the rate of exponential growth being k = 1, which corresponds to [Ag] = e^1^ * ln[Ag]. Shape parameter d of the generalized logistic function allows for asymmetry in the binding curve. In our assay, d is an index, which characterizes the Ab composition of serum. The location of the fastest growth x_i_ with respect to ln [Ag] provides a general measure of antibody affinity.

Affinities and concentrations of serum antibodies are distributed over a very wide range, equilibrium dissociation constants from 10^−5^ to 10^−11^ and molar concentrations from pM to nM, respectively. In order to better assess interactions over this range and to weigh curve fitting against signal-intensity-dependent variation, we use logarithmically transformed signals of binding. Thus, fitting the Richards curve to our measurements therefore requires the logarithmic form of the above equation:(5)$$\mathrm{lnR}\left(x\right)=\ln\;\left(A\right)+\frac{1}{1-d}*\ln\left(1+\left(d-1\right){\;*\;e}^{-k*\left(x-x_{i}\right)}\right)$$

By using the logarithm, we transform thermodynamic activity to chemical potential of the Ab, as determined by its standard molar Gibbs free energy, concentration and activity coefficient:(6)$$\mu_{\mathrm{Ab}}=\mu {}^{\circ}+\mathrm{RTln}\left(a_{\mathrm{Ab}}\right)=\;\mu {}^{\circ}+\mathrm{RTln}\left(c_{\mathrm{Ab}}{\;*\;\gamma }_{\mathrm{Ab}}\right)=\mu {}^{\circ}+\mathrm{RTln}\left(c_{\mathrm{Ab}}\right)+\mathrm{RTln}\left(\gamma_{\mathrm{Ab}}\right)$$
where μ is chemical potential; μ° is thestandard chemical potential or molar Gibbs free energy; a is relative thermodynamic activity; γ is the activity coefficient; and c is the molar concentration.

Since the standard term serves as a reference point, we can still obtain relative potentials after its removal. Then, we can interpret ln(A) of the lnR(x) function as the mole fraction term of chemical potential, and the remaining part of the function describes the activity coefficient as a function of [Ag] (for details see Appendix A).

To demonstrate the dependence of ln(A) from the serum dilution we extended the formula of lnR(x) to obtain the combined fitting of serum dilutions and antigen dilutions:(7)$${\mathrm{lnR}}_{2}\left(x,z\right)=\ln\;\left(C\right)-m*\ln\;\left(B+z\right)+\frac{1}{1-d}*\ln\left(1+\left(d-1\right){\;*\;e}^{-\left(x-x_{i}\right)}\right)$$
where z is the serum dilution, and m, B, C are auxiliary parameters of the logarithm of another generalized logistic function [29] of −ln(z) that replaced ln(A) and has a parametrization different from Formula (5).

Reliable curve fitting depends on the number and locations of x = ln[Ag] measurement points on the curve and requires the value of x_i_ to be in the measurement range.

### 3.3. Characterization of Anti-Deamidated Gliadin Peptide Serum Antibodies

As an exemplary antigen for the proof of our measurement principle we choose a deamidated peptide sequence known to be the target of antibodies in celiac disease. Such an antigenic peptide is a well-defined molecular target of both IgA and IgG antibodies in celiac patients and is used in laboratory diagnostics. We used peptide concentrations corresponding to the range of K_D_ values expected to occur in serum and peptide dilutions that extended well beyond that range. Serum samples were diluted to span about two orders of magnitude and correspond to dilutions conventionally used in serological diagnostics. Binding data were fitted using the lnR_2_(x) function of Equation (7) introduced above and generated curves were overlain on the binding data (Figure 2).

Using the values of parameters from the fitted binding curves of experimental data, we can generate binding curves normalized to Ab concentrations, which are curves of thermodynamic activity coefficients $\gamma_{\mathrm{Ab}}$, as illustrated in Figure 3. These curves are independent of antibody concentrations and fluorescent intensity and are therefore comparable for different classes of Ab in a given sample, for different samples and any combination thereof. Parameter x_i_ and $\gamma_{\mathrm{Ab}}$ are sufficient to quantitatively characterize the distribution of Ab thermodynamic activity in the tested serum sample as long as the Richards curve models binding events.

To confirm that conventional titration cannot provide exact results we calculated titers from the binding data obtained from the microspot experiments. The classical approach of serum titration was heavily dependent on the antigen density, as observed by others [13,30,31]. We calculated end-point titers and mid-point titers of the same measurements by logistic fitting of binding curves to distinct antigen densities and obtained different titers for different antigen densities (Figure 4). This antigen density dependence is not only avoided but is instead exploited in our approach of fitting curves to antigen density.

We also confirmed the applicability of our method to other antigens using a microarray printed with dilution series of citrullinated peptides known to be targets of autoantibodies of rheumatoid arthritis patients (Appendix A).

### 3.4. Measurement of Reference Monoclonal Antibody Properties

Current serological assays use standardized antibody preparations as reference material for quality control and calibration [10]. These preparations are monoclonal antibodies alone or in combinations of monoclonal antibodies, or pooled raw or isolated serum antibodies from positive samples. Because of the individual variance in the affinity distribution of serum antibodies, these approaches have the major drawback of assuming identical affinity distribution of reference and test samples. Titration by antigen density avoids this pitfall since it does not a priori assume, but it rather identifies distribution of samples and reference standards. The incorporation of a reference antibody is also suitable to control antigen density variation. A critical point of antigen density titration is the precise deposition of Ag on the microarray surface, so that the real density of antigen corresponds to the nominal antigen concentration used for printing. The precision can be estimated by measuring and characterizing a reference antibody. We successfully used the monoclonal antibody 14D15 specific for an epitope in our deamidated gliadin peptide sequence to characterize their interaction (Figure 5).

## 4. Discussion

The concentrations of serum antibodies and of targets of these antibodies, along with the range of the strength of these interactions span several orders of magnitude. To map these interactions we need technologies that provide quantitative results over this wide range, from low picomolar to micromolar concentrations. Here we present results, which together with earlier observations [32,33] provide evidence and theoretical support for protein microspot-based fluorescent detection as a method of choice.

The difficulty of rendering the measurement of polyclonal antibody responses quantitative lies partly in the reproducible generation of rigorously characterized antigen, partly in the mathematical description of heterogeneity of polyclonal antibody response. The generation of synthetic epitopes for serological assays, along with functional characterization using reference affinity reagents may solve the first issue [10,34]. Our approach may provide a solution for the second one: establishing standardized assays that generate quantitative data with universal biochemical units of measurement. The key for generating a quantitative description of polyclonal serum antibody reactivity is the ability to dissect antibody concentration and affinity, the two parameters that determine antigen saturation. This is basically achieved by titrating both concentration and affinity: in addition to the conventional serum dilution series, a series of antigen dilutions is used concomitantly. By diluting serum antibody, a series of antibody concentrations is examined, while by additionally using microspots of antigen dilution series, affinity distribution is assessed simultaneously. Microspots allow a mass-independent interrogation of interactions on the chip surface, whereby the measurement becomes independent of the relative total masses of Ag and Ab in the reaction volume. Complex formation is determined only by concentration and affinity, which two measures can be deconvolved by two-dimensional titration. By gradually increasing the Ag density of the surface, an Ab with a given affinity will engage in lower affinity interactions as well. This results in the measurement of a range of affinities from the highest (detected on low density Ag spots) to more degenerate lower affinity interactions.

There have been several attempts to characterize the affinity of antibodies associated with autoimmunity [35,36,37,38], infectious disease [39,40], vaccination [41] and allergy [14,42,43,44,45]. The strength of binding of serum antibodies is determined by the distribution of clones with different affinities specific for the tested epitope. While the importance of antibody affinity in conferring pathogenicity or protectivity is widely acknowledged, affinity determination is not generally considered as part of diagnostic serology. The possible reason is that affinity measurement is technologically challenging and is not standardized, as outlined above. Affinity dependence of traditional immunoassays (e.g., RIA, ELISA, Farr assay, hemagglutination, complement-mediated haemolysis, precipitation) [13,15,31,46] practically excludes low affinity interactions from measurement and may skew affinity determinations when such assays are used. Such a progressive decrease of sensitivity to low affinity antibodies should lead to estimations of antibody affinity distribution with a higher mean affinity. The method we propose here presumes the existence of low affinity antibodies and measures them by using Ag at very high densities as well.

Microspot immunoassays were introduced for the measurement of analyte concentration, taking advantage of the very low amount of capture antibody required by the technology. If in the complete volume of measurement, the concentration of antibody on the solid support is less than 1% of the K_D_ of interaction, then the concentration of the analyte does not change significantly (<1%) during the measurement. The signal from a detection antibody is therefore correlated only with analyte concentration when other variables are kept constant, which conditions are characteristic of ambient analyte immunoassay [19]. The analyte in our assay is serum antibody and the K_D_ of interactions are heterogeneous. Nevertheless, the amount of antigen and therefore its concentration in the total measurement volume is still negligible. Our method takes further advantage of microspots by varying the surface density of antigen for antibody capture. Since the relative concentrations of antibodies with different affinities are not changed by serum dilution, the affinity profile of antibodies that make contact with antigen on the solid support does not change either. Identical saturation of the antigen spots would give a linear increase of fluorescent signals in Figure 3A,B. The observed non-linearity indicates changing degrees of antigen saturation at a given serum antibody concentration. In physical chemistry the thermodynamic activity coefficient adjusts concentrations to effective concentrations, accounting for non-ideality of binding. Therefore, instead of fractional occupancy we use thermodynamic activity coefficient. Dependence of the activity coefficient of serum antibodies on the density of antigen [Ag] relative to the average affinity of interactions is modelled by the Richards function. By varying [Ag] we obtain a distribution curve of activity coefficients suitable for estimating values of the parameters of fitted function.

The interpretation of Richards curve parameters provides insight into several aspects of serum antibody reactivity. Parameter A is related to the molar concentration of functional antibody binding sites, called paratopes, belonging to the measured immunoglobulin class. Because the measurement of distribution of affinity reaches into the low affinity range, the estimated concentrations are approximations of total serum antibody concentrations. We expect this value to be dependent and correlated with serum antibody concentrations and therefore to have less diagnostic value. Thus, it is presumably not necessary to adjust for antibody valency and convert binding site concentrations into isotype concentrations. Antibody responses that exploit the increase in antibody concentrations without major increase in affinity, such as the thymus-independent response, are therefore expected to appear as a change in this parameter. A technical advantage of rendering the binding parameters independent of total serum antibody concentrations is that we also exclude absolute fluorescence intensity values from the analysis, thus removing a factor of interlaboratory variance.

The central point or inflection point x_i_ is related to the apparent standard chemical potential of serum antibodies. The lower the x_i_, the stronger the binding, similar to Ab-titration-based affinity determination approaches [15]. We propose to use the short name “lnK_D_” for the natural logarithm of molar concentration of antigen, x_i_, required to reach maximal relative growth of antibody activity coefficient γ_Ab_ (inflection point of sigmoid curve in the log-lin scale). While the unit of chemical potential is that of energy (Joule/mole) here we would retain the unitless number derived from antigen concentration titration.

The Richards curve uses parameter d to introduce asymmetry on the two sides of the inflection point. At d = 2, the curve is symmetric, and it is a logistic curve. At d → 1, the curve approaches the Gompertz growth curve and becomes asymmetric as the inflection point shifts from y_i_ = A/2 to y_i_ = A/e. The slope of the lnR_2_(x) function s = k/(d − 1) at minus infinity, in the case k = 1, meanwhile increases from s = 1 to ∞. This slope characterizes interactions as the antigen is diluted out to infinity. Infinite dilution is a special thermodynamic state when antigen molecules are in contact with antibody molecules only [47]. In this state, binding is determined only by interactions between Ag and Ab, without the interference of homotypic interactions. This ideal state is characterized by the limiting activity coefficient $\gamma_{i}^{\infty }$. Using our parametrization d-1 changed between 0 and 1. We propose that d − 1 is related to the limiting activity coefficient of Ab, $\gamma_{\mathrm{Ab}}^{\infty }$, which characterizes the composition of the Ab mixture. We expect that parameter d can be used to characterize disease activity when antibody diversity and affinity is related to disease pathogenesis, with lower d implying immunological activity and higher d indicating the approach of equilibrium concentrations and stability of the immune response.

A potential disadvantage of the proposed method is that it is technologically challenging compared to established automated assays. A critical point is the generation of antigen spots with real densities as close to nominal densities as possible. This requires the introduction of extra quality control steps into the production. The fitting procedure requires a minimal number of data points around the estimated inflection point; therefore, weakly reactive samples cannot be quantitated. These disadvantages position our assay as a fine analysis method, suitable not for the screening and identification of presence of specific antibodies but rather for the detailed characterization of pre-screened sera. The binding and detection of serum antibodies is a complex process, with various classes and clones of antibodies competing for binding, engaging with varying valencies, these subsequently being bound again by secondary antibodies. While we cannot pinpoint the contribution of each of these factors to the emergent properties of the system, we propose that fitting the Richards curve is a reasonable and achievable approach for the quantitative characterization of these events.

A key conceptual novelty in our assay is that instead of assessing the concentration of a heterogeneous antibody population, we assess the distribution of apparent chemical potential. Current immunoassays focus on establishing conditions that are ideal for the estimation of the concentrations of antibodies against a particular target antigen. For solid phase assays, this involves the optimization of coating antigen density, where an antigen density suitable for identifying the shift of average K_D_ in a diagnostically relevant region of affinity is actually identified. By using microspots with antigen densities spanning the whole range of relevant concentrations, a single measurement can provide a distribution of apparent chemical potentials.

The procedure we describe here assumes that equilibrium is reached when microspots are incubated in serum. In order to confirm that measurement conditions are appropriate and to allow reproducibility and interassay comparisons, antibody standards can be introduced into the assay. Pooled positive and borderline serum samples could be used in our assay just as in any other immunoassay for specific antibody measurement. Monoclonal antibody preparations are also suitable as internal reference, but unlike current practices where simply monoclonal binding signals are used for normalization [48], binding parameters obtained by curve fitting would be used for quality control or data adjustment.

## 5. Conclusions

The antigen microspot titration assay we introduce in this paper generates biochemical units for the characterization of specific antibody. Because of the universal units, the comparison of serological reactivity against different antigens, mediated by distinct antibody isotypes, as demonstrated here for IgG and IgA, becomes possible. This is a key advantage compared to current serological assays using arbitrary units or titers. Therefore, the introduction of such quantitative assays could not only improve diagnostic accuracy of immunoassays but would also pave the way to our quantitative understanding of the adaptive immune system. The generalized quantitative model of antibody homeostasis [3] provides a conceptual framework for the experimental mapping of the “antibodyome” [49,50,51], a goal that could be achieved by the strategic mapping of antigen structural space. The integration of quantitative binding information into a network and the mapping of the interaction network [50] to the sequence space generated by next generation sequencing is expected to bring about unprecedented systemic understanding of adaptive immunity in particular and protein evolution in general.

## 6. Patents

A provisional patent related to this work has been filed by Diagnosticum Zrt.

## Figures and Tables

**Figure 1 sensors-22-03962-f001:**
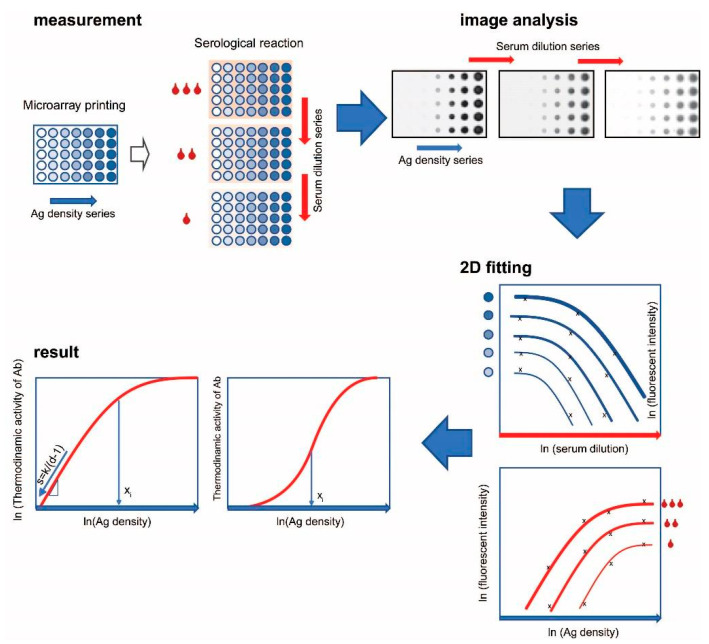
Simultaneous titration of antigen density and serum antibody. Steps of the technology, starting with microarray fabrication and measurement, through image analysis to curve fitting and visualization of results are shown. Serum dilution is indicated by red drops, antigen density differences are represented by shades of blue circles. Several binding curves are transformed into one by fitting data with generalized logistic curves, yielding two parameters, x_i_ and d, which characterize thermodynamic activity distribution of serum antibodies. Affinity is related to x_i_, the point of inflection; antibody heterogeneity is related to d, the asymmetry parameter. The slope at infinity “s” of the curve in the lower left corner is given by the equation shown.

**Figure 2 sensors-22-03962-f002:**
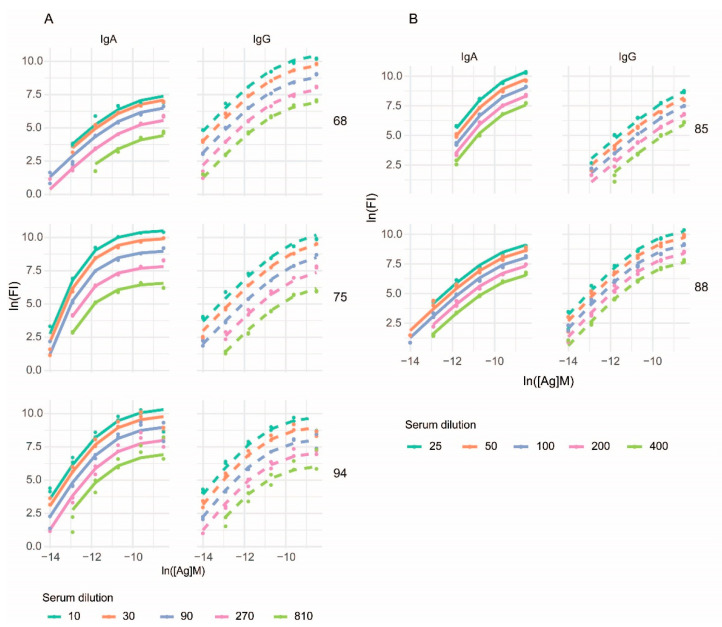
Fitting in two dimensions. Examples of parallel measurements of IgG and IgA binding to diagnostic celiac disease peptide epitope are shown, with serum dilutions in (**A**) three-fold and (**B**) two-fold steps, as shown beside the panels, in five representative serum samples. Dot symbols stand for measurement data, lines are fitted curves.

**Figure 3 sensors-22-03962-f003:**
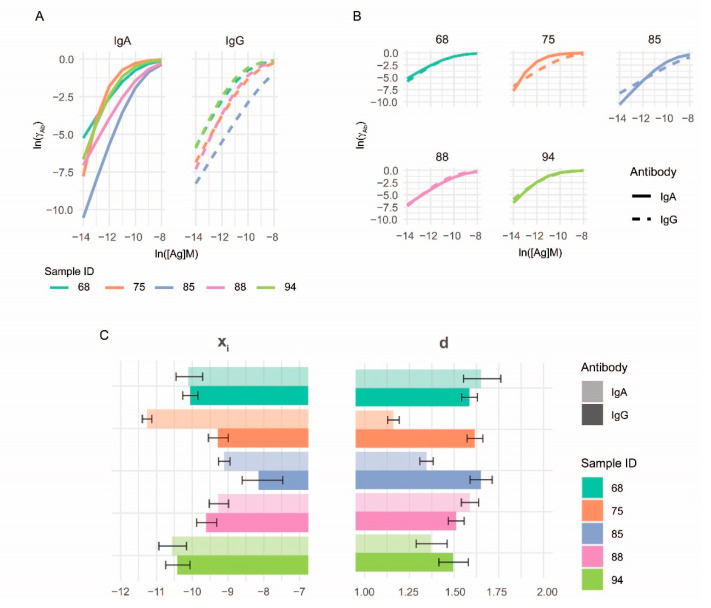
Comparative characterization of serum antibody binding. Using the values of the parameters obtained by curve fitting, calculated thermodynamic activity coefficient distributions are comparable not only for (**A**) serum samples but also for (**B**) distinct antibody classes. Results shown in Figure 2 were used for the generation of normalized comparisons. Bar charts (**C**) show mean d and x_i_ values, and 95% confidence intervals as whiskers.

**Figure 4 sensors-22-03962-f004:**
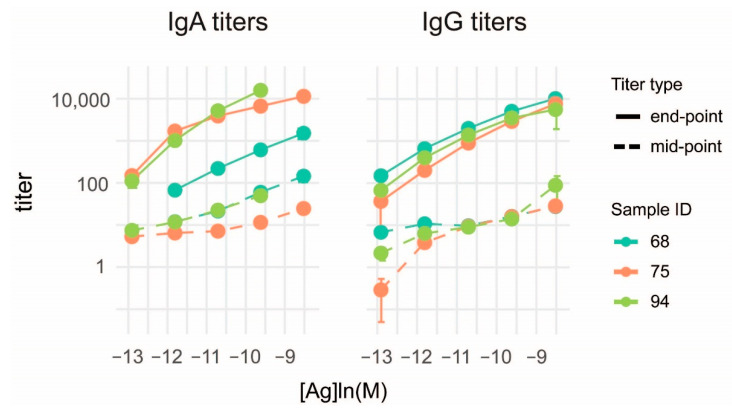
Conventional titration of serum. Mid-point and end-point titers of IgA and IgG are calculated by logistic fitting of binding to distinct antigen densities and are displayed as a function of antigen density. Logarithm of the titers at 5 tested antigen densities are represented by filled circles, connecting lines are linear interpolations. Error bars represent standard deviation.

**Figure 5 sensors-22-03962-f005:**
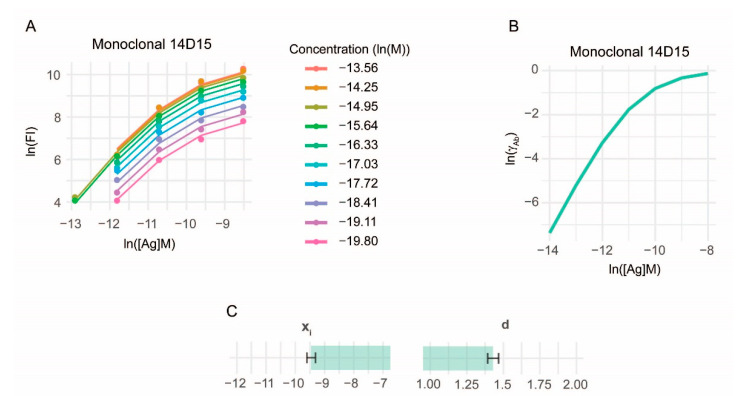
Characterization of a monoclonal Ab for reference. Binding data of monoclonal Ab 14D15 was fitted using the same algorithm as for serum antibodies. Binding data and fitted curves (**A**), calculated distribution of activity coefficient (**B**) and estimates of parameters x_i_ and d (**C**) are shown.

## Data Availability

Experimental data are available upon request.

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
