# Peer review of "Absolute Quantitation of Serum Antibody Reactivity Using the Richards Growth Model for Antigen Microspot Titration"

_sensors, 2022, doi:10.3390/s22103962_

Round 1

Reviewer 1 Report

 The authors proposed novel method to quantify polyclonal antibody reactivity relies on the simultaneous titration of immobilized antigen and serum dilution and on an improved mathematical model of fitting. There are some points that can be changed to improve the quality of the manuscript

1)       Line 108: What buffer was used for peptide printing?

2)       Line 118: Please, move information about circular citrullinated peptide in Supplementary figure S1 as they are not mentioned anywhere in the main text

3)       Line 118: Please, add code numbers and manufacturers for “commercially available ELISA kits” Why ELISA kits for transglutaminase antibodies not for gliadin antibodies were used?

4)       Line 126: Please, add dilution steps for human sera

5)       Line 199: “The solid surface used as antigen adsorbent has higher binding capacity than conventional polystyrene” – Any proof?

6)       Line 131: Please, check if this is correct “(Jackson Immunoresearch, Ref.: 109-496-008)”. An antibody with this code number cannot be found in the manufacturer's catalog

7)       Line 136: Section “Measurements with 14D5 monoclonal antibody”:

a.       Did the authors use double-blocked slides for this protocol? The slides were blocked in 0.1 M Tris buffer (pH=8,0) for 1h at 37°C (line 110) and then blocked in blocking buffer (0.05% Tween 20, 2% BSA, PBS) for 30 minutes at 37°C (line 139). But the authors didn’t use the second blocking step for human samples, why?

b.       Please unify “(PBS, 2% BSA, 0.05% Tween 20)’’ and “(0.05% Tween 20, 2% BSA, PBS)” throughout the Materials and Methods Section.

c.        Please, add incubation times.

8)       Line 142: Please, add dilution/concentration for goat anti-mouse IgG2

9)       Line 164: “It is the nature of our measurement system that the logarithm fluorescent intensity is a linear function of the logarithm concentration of AbAg complexes”

a.       It can be true only for a linear range of measured concentrations.

b.       Dilution linearity is a great problem for measuring autoantibodies. Some specimens may not dilute linearly because of the heterogeneity of the autoantibodies with respect to physiochemical properties. Even commercial kits for medical use didn’t overcome this phenomenon. See, for example, Section “LIMITATIONS OF THE PROCEDURE'' for Architect Anti-TPO Reagent Kit 2K4720, (Abbott Laboratories, USA) (https://www.ilexmedical.com/files/PDF/Anti-TPO_ARC.pdf) and section “Dilution” for Anti-Tg Elecsys, cobas е #06368697190 (Roche Diagnostics, Germany) (https://labogids.sintmaria.be/sites/default/files/files/anti-tg_2018-07_v6.pdf). “The autoantibodies are heterogeneous and this gives rise to non‑linear dilution phenomena”

10)   Line 214, figure 1 caption:

a.       Add description in the text not several curves are transformed into one (2D fitting-result)

b.       Please, give some explanation for parameters “xi and d” in the text

c.        Please, add comment about S=k/(1-d) in the text

d.       Please explain how “xi and d” relate to the affinity of the antibodies

11)   Line 229: What is the evidence for equilibrium?

12)   Line 297, figure 2 caption: Serum dilution 25, 50, 100, 200, 400. There are “serum dilutions in (A) three-fold and (B) two-fold steps”?

13)   Line 325, figure 4 caption: What namely were titers for these experiments?

14)   Line 357 Please, phrase these statements. “To obtain a systems level structured map of these interactions we need technologies that provide quantitative results of molecular interaction measurements over this wide range, from low picomolar to micromolar concentrations. Here we present results, which together with earlier observations [32,33] provide evidence and theoretical support for protein microspot-based fluorescent detection as a method of choice”. There is no information about autoantibodies concentrations in present article, add it, please

15)   Line 356“The difficulty of rendering the measurement of polyclonal antibody responses quantitative lies partly in the reproducible generation of rigorously characterized antigen, partly in the mathematical description of heterogeneity of polyclonal antibody response.” – Measurement difficulty lies in the mathematical description of heterogeneity, not on heterogeneity?

16)   Line 364Our approach may provide a solution for the second one: establishing standardized assays that generate quantitative data with SI units of measurement”. – Where are these units in the present article? Add information, please

17)   Line 388. Please add references for these statement “Affinity dependence of traditional immunoassays (e.g., RIA, ELISA, Farr assay, hemagglutination, complement-mediated haemolysis, precipitation) [46] practically excludes low affinity interactions from measurement and may skew affinity determinations when such assays are used”

18)   Line 391: What results in this article can demonstrate this statement? “The method we propose here presumes the existence of low affinity antibodies and measures them by using Ag at very high densities as well.”

19)   Line 408: What do the authors mean by “the activity coefficient… of antibodies”?

20)   Line 477: Please, add reference here: “current practices where simple monoclonal binding signals are used for normalization”

21)   Line 492: Please, add Supplementary text 2

22)   Symbols and abbreviations: What do the authors mean by “antibody capacity [FI]”?

Reviewer 2 Report

This manuscript presents a methodology for quantification of serum antibody reactivity based on a Richards growth model applied to antigen microspot titration. The approach may provide some advantages for more consistent characterisation of serum antibody binding properties. The manuscript is well-presented and appropriately laid out with data summarised in a useful set of figures. This manuscript is a resubmitted version of an earlier manuscript and in the resubmission the authors have adequately addressed points raised in my earlier review. There are some points for the authors to consider and these are detailed below.

  1. Page 3 line 118 “. . . with commercially available ELISA kits.” Can you please give succinct details in the manuscript on the type of kits and the suppliers.
  2. Page 4 lines 149-150 “. . . using 80% of the diameter . . . by the software.” Succinctly explain in the manuscript why you settled on 80% of the diameter. Is this to avoid edge effects?
  3. Page 5 lines 199-200 “The solid surface . . . conventional polystyrene.” Can you provide a supporting reference for this statement?
  4. Page 7 line 292 “Serum samples were diluted . . .” In the manuscript, clarify the diluent used for the serum sample dilutions.
  5. Figure 4. Add error bars to the plots in this figure.

Round 2

Reviewer 1 Report

The authors have answered all the reviewers’ comments and modified the manuscript accordingly.

Minor remarks:

1) Affiliation 6 - the authors?

2) At first mention of a manufacturer, the town (and state if USA) and country should be provided

3) Letters, numbers and signs in figures 2, 3, 5 must be clear

4) Conclusions: In this final part, the authors did not conclude about the assays they have described here. They have stated ideas about “strategic mapping of antigen structural space” or others but I consider that they have to obtain some conclusions about how the technology allows “quantitative comparison of different samples, isotypes or antigens” as they have stated in Introduction.

Author Response

Dear Reviewer,

Thank you again for your remarks and suggestions. We have corrected and supplemented the text as detailed below.

1, The mark of the 6th affiliation was added to the appropriate author.

2, Manufacturers' towns and states are now given.

3, The size of figues was increased to improve readability. We uploaded vector graphical images for the published version, which will also improve quality.

4, We added sentences to Conclusions to summarize not only theoretical but also practical advantages of a quantitative serological assay.

regards,

on behalf of all the authors

This manuscript is a resubmission of an earlier submission. The following is a list of the peer review reports and author responses from that submission.

Round 1

Reviewer 1 Report

This manuscript reports an analytical approach to the characterisation of the binding between serum antibodies and immobilised antigens in a microspot titration method. The basis for this approach lies in serially diluting both the raw serum antibody and the spotted antigen concentrations and then applying 2D fitting to the serum dilution and antigen density data and combining them to produce a log/log plot from which two key useful parameters are obtained: a point of inflection and an asymmetry parameter. The paper is generally well written with a suitable set of supporting figures. The approach taken may provide advantages in better and more consistently characterising the binding properties of serum antibodies. There are some points that I feel the authors should address and these are detailed below.

  1. Page 3 line 115. “Previously characterised . . .” This implies that the serum samples have been subject to some previously reported characterisation. Clarify the characterisation that has been performed previously and provide a reference/s where appropriate.
  2. In the materials and methods section, the authors should include details of where the monoclonal antibody against deamidated gliadin (14D5) comes from. You should do likewise for the IgG and IgA.
  3. Page 3 line 133 “The previously described basic protocol . . .” Provide a supporting reference for this protocol.
  4. Page 4 lines 146-147 “. . . and values of the . . . all RFI values . . .” Acceptable use of the lowest antigen concentration as a background reference is critically dependent upon there being negligible binding at this concentration. Given, you don’t know the binding properties of a given serum antibody in advance, how can you know which minimum concentration to use and whether it would actually be suitable as a background reference?
  5. Page 4 lines 162-163 “Formula (6) also implies . . .” Referring to formula (6) here is jumping a long way ahead, as formula (6) is not explained/defined until the results section. The formula should really be defined before referring to it in the text.
  6. Figure 3C. The labelling of the parameters xi and d needs to be larger and more legible.
  7. Figure 4. You need to make it clearer that the mid-point data are the dashed lines on the plots.
  8. Page 13 lines 459-462 “Additionally, monoclonal antibody . . . for adjusting calculations.” More detail is needed about how you see monoclonal antibodies serving as internal references. What do you mean by ‘controls for validation’ and ‘adjusting calculations’? How exactly might monoclonal antibodies be used alongside serum antibodies to improve characterisation of serum antibody binding?

Reviewer 2 Report

The problem that the authors are trying to solve is of great interest. Considering the specifics of the analysis of autoantibodies, it should be noted that the detected autoantibodies consist of a polyclonal mixture that differs in affinity for the autoantigen. Moreover, autoantibodies can be represented by immunoglobulins of various isotypes and subclasses, for which there is a significant variety of targets - autoantigens, even in one autoimmune disease. Proposed novel method to quantify polyclonal antibody reactivity relies on the simultaneous titration of immobilized antigen and serum dilution and on an improved mathematical model of fitting. A very attractive idea, but the experimental part has major concerns.

Line 89. Please, clarify this statement “When affinity is constant and homogenous, as in a capture immunoassay utilizing monoclonal antibody, the density of analyte bound to the surface is determined solely by its concentration.” What do the authors mean by “analyte bound to the surface”?

  1. Materials and Methods. The entire experimental part should be described in more detail. Including manufacturers or method of obtaining for detection antibodies, immobilized peptide.

Line 126 It may be better to use “fluorescently labeled antibodies” instead of fluorescently labelled antibodies”

Line 133 A reference to the protocol described earlier is needed here. “The previously described basic protocol was altered for the assays with monoclonal antibody against deamidated gliadin (14D5).”

For 14D5 antibodies and anti-species antibodies, the manufacturer or method of preparation must be specified.

The authors used the following dyes and antibodies (Ex/Em, nm): anti-igG-DL 649 (646/672), anti-igA-DL 549 (550/568). It’s not clear from the Materials and Methods section was it the mix of antibodies in one assay? What concentrations were used? Why was a third dye chosen for experiments with 14D5 (goat anti-mouse IgG2a-Alexa488 (493/ 518))? What microarray scanner was used to obtain fluorescent images?

Figure 1. On the left side, the microarray should contain 5 spots each. Maybe it’s better to draw microarray layout as in Material and method the authors wrote “Different peptide dilutions, as indicated in the text, were spotted in quintuplicates of ½ serial dilution in 7 steps starting from 200 μM.” I couldn’t find these seven dilutions in Figure 1. In figure 1 for only 4 peptide dilutions the resulting signals after assay are above background. Has the experimental saturation of the signal intensity been achieved? For seven dilutions, why not use a higher concentration to start with?

Is there any repeat in the sample assay? What serum dilutions were used? Different for different samples?

Supplementary figure S1. The information about the assay is needed. What citrullinated antigen was used? All samples were positive for both IgG ACPA and IgM ACPA? Do these samples contain rheumatoid factor? Any controls, healthy donors?

Some facts that can affect results obtained with the proposed method can be breathily discussed. Even if we assume that the steric hindrance and a heterogeneity due to immobilization are absent. However, the measured antibodies of different classes essentially compete with each other for binding to the immobilized peptide. If there are other antibodies (of a different class) in the sample, they will affect the interaction, igM, for example. While one igG molecule can theoretically interact with both one peptide molecule and two. Serum igA can be either a monomer or a dimer. IgM have multiple binding sites that recognize the same epitope. Non-specific binding can occur in some samples. Detecting antibodies are also essentially a mixture of antibodies with a different number of dye molecules on each, including also some proportion of unlabeled proteins. So, these are multiple polyvalent interactions that differ in each sample. That's why any mathematical model should be supported by very careful experimental work.

Reviewer 3 Report

The study by Papp and co-workers aims for the quantitation of serum antibody reactivity. There are some points that could be included/modified to further improve the quality of the manuscript.

1.Figures 2 to 5 need to be redrawn as clear as Figure 1, the current presentation of these figures does not look stringent and scientific.

2.What is the source of Equation 1? It is not mentioned in references 24 and 25. Also, is there any relationship between Equation 1 and Equation 2?

3.I don't quite understand how xi (value of x at inflection point) and d (shape parameter) values relate to the activity of the antibodies.

4.There is only qualitative analysis throughout the text, and there should be quantitative analysis.

5.I suggest that the authors should find other methods to compare your method to increase the validation of these experiments.